# Morning exercise and pre-breakfast metformin interact to reduce glycaemia in people with type 2 diabetes: a randomized crossover trial

Brenda J. Peña Carrillo[1,2] , Emily Cope[1,2], Sati Gurel[2], Andres Traslosheros[2], Amber Kenny[1,2], Oscar Michot-Duval[2], Nimesh Mody[1], Mirela Delibegovic[1], Sam Philip[3], Frank Thies[2], Dimitra Blana[4] and Brendan M. Gabriel[1,2,5]

[1] *Aberdeen Cardiovascular and Diabetes Centre, School of Medicine, Medical Sciences & Nutrition, University of Aberdeen, Aberdeen, UK*
[2] *The Rowett Institute, School of Medicine, Medical Sciences & Nutrition, University of Aberdeen, Aberdeen, UK*
[3] *NHS Grampian Diabetes Research Unit, Diabetes Centre, Aberdeen Royal Infirmary, Aberdeen, UK*
[4] *Centre for Health Data Science, Institute of Applied Health Sciences, University of Aberdeen, Aberdeen, UK*
[5] *Department of Physiology and Pharmacology, Integrative Physiology, The Karolinska Institute, Stockholm, Sweden*

**The Journal of Physiology**

Handling Editors: Karyn Hamilton & Josiane Broussard

The peer review history is available in the Supporting Information section of this article (https://doi.org/10.1113/JP285722#support-information-section).

**Abstract** Exercise is recommended in the treatment of type 2 diabetes and can improve insulin sensitivity. However, previous evidence suggests that exercise at different times of the day in people with type 2 diabetes may have opposing outcomes on glycaemia. Metformin is the most commonly prescribed initial pharmacological intervention in type 2 diabetes, and may alter adaptions to exercise. It is unknown if there is an interaction between metformin and diurnal exercise outcomes. We aimed to investigate glycaemic outcomes of moderate intensity morning *vs.* evening exercise in people with type 2 diabetes being prescribed metformin monotherapy. In this study, nine males

This article was first published as a preprint. Peña Carrillo BJ, Cope E, Gurel S, Traslosheros A, Kenny A, Mody N, Delibegovic M, Philip S, Thies F, Blana D, Gabriel BM. 2023. Morning exercise and pre-breakfast metformin interact to reduce glycaemia in people with Type 2 Diabetes: a randomized crossover trial. medRxiv. https://doi.org/10.1101/2023.09.07.23295059

and nine females with type 2 diabetes undergoing metformin monotherapy (age 61 ± 8.2 years, mean ± SD) completed a 16-week crossover trial including 2-week baseline recording, 6 weeks randomly assigned to a morning exercise (07.00–10.00 h) or evening exercise (16.00–19.00 h) and a 2-week wash-out period. Exercise arms consisted of 30 min of walking at 70% of estimated max heart rate every other day. Glucose levels were measured with continuous glucose monitors and activity measured by wrist-worn monitors. Food-intake was recorded by 4-day food diaries during baseline, first and last 2 weeks of each exercise arm. There was no difference in exercise intensity, total caloric intake or total physical activity between morning and evening arms. As primary outcomes, acute (24 h) glucose area under the curve (AUC), was lower ($P = 0.02$) after acute morning exercise (180.6 ± 68.4 mmol/l) compared to baseline (210.3 ± 76.7 mmol/l); and there were no differences identified for glucose (mmol/l) between baseline, morning and evening exercise at any specific time point when data were analysed with two-way ANOVA. As secondary outcomes, acute glucose AUC was significantly lower ($P = 0.01$) in participants taking metformin before breakfast (152.5 ± 29.95 mmol/l) compared with participants taking metformin after breakfast (227.2 ± 61.51 mmol/l) only during the morning exercise arm; and during weeks 5–6 of the exercise protocol, glucose AUC was significantly lower ($P = 0.04$) for participants taking metformin before breakfast (168.8 ± 15.8 mmol/l), rather than after breakfast (224.5 ± 52.0 mmol/l), only during morning exercise. Our data reveal morning moderate exercise acutely lowers glucose levels in people with type 2 diabetes being prescribed metformin. This difference appears to be driven by individuals that consumed metformin prior to breakfast rather than after breakfast. This beneficial effect upon glucose levels of combined morning exercise and pre-breakfast metformin persisted through the final 2 weeks of the trial. Our findings suggest that morning moderate intensity exercise combined with pre-breakfast metformin intake may benefit the management of glycaemia in people with type 2 diabetes.

(Received 25 September 2023; accepted after revision 29 February 2024; first published online 28 March 2024)

**Corresponding author** B. M. Gabriel: The Rowett Institute, University of Aberdeen, Ashgrove Rd W, Aberdeen AB25 2ZD, UK.    Email: Brendan.gabriel1@abdn.ac.uk

**Abstract figure legend** Morning exercise and pre-breakfast metformin interact to reduce glycaemia in people with type 2 diabetes.

## Key points

- Morning moderate exercise acutely lowers glucose levels in people with type 2 diabetes being prescribed metformin.
- This difference appears to be driven by individuals that consumed metformin prior to breakfast rather than after breakfast.
- Morning exercise combined with pre-breakfast metformin persistently reduced glucose compared to morning exercise combined with post-breakfast metformin through the final week (week 6) of the intervention.
- Our study suggests it may be possible to make simple changes to the time that people with type 2 diabetes take metformin and perform exercise to improve their blood glucose.

## Introduction

Type 2 diabetes mellitus is a growing, global health challenge (Saeedi et al., 2019). Exercise is often recommended by front-line clinicians as a potent therapeutic treatment in people with type 2 diabetes, and it can improve insulin sensitivity (Gabriel & Zierath, 2017). However, the optimal time of day to perform exercise for people with type 2 diabetes has not been fully elucidated. Previous evidence suggests that morning high intensity interval training over 2 weeks increases blood glucose levels in people with type 2 diabetes (Savikj et al., 2019), drawing parallels to the dawn phenomenon. Conversely, evening exercise reduced glycaemia on the

day of exercise in people with type 2 diabetes (Savikj et al., 2019). This time-of-day of exercise effect upon blood glucose regulation appears to be supported by other recent studies (Mancilla et al., 2021; Moholdt et al., 2021; Qian et al., 2023; van der Velde et al., 2023). These apparently divergent outcomes of morning *vs.* evening exercise highlight that it is imperative to optimize the timing of exercise for people with type 2 diabetes.

Several other studies have assessed the time-of-day effect of exercise upon glycaemic or insulin-related outcomes. For example, cross-sectional data in 775 participants demonstrates that moderate-to-vigorous exercise in the afternoon or evening was associated with a greater reduction of insulin resistance compared with an evenly distributed daily time pattern of exercise (van der Velde et al., 2023). Further associative evidence (Qian et al., 2023), demonstrates that in 2416 participants moderate-to-vigorous physical activity performed in the afternoon is associated with improvements in glycaemic control in people with diabetes. In a randomized trial, men who were overweight or obese demonstrated improvements in glycaemic control after evening exercise, while a group who exercised in the morning saw no improvements in glycaemic control (Moholdt et al., 2021). In another randomized trial assessing people with compromised metabolism, a group that performed exercise training in the evening experienced superior beneficial improvements in fasting plasma glucose levels compared to a group that performed exercise training in the morning (Mancilla et al., 2021). However, the time-of-day effect of exercise on glycaemic related outcomes may not be consistent across all cohorts.

Skeletal muscle is key in the metabolic response to exercise (Gabriel & Zierath, 2017) and it has been demonstrated that people with type 2 diabetes have an intrinsically altered circadian rhythm in skeletal muscle, at the level of both the molecular clock and mitochondrial function (Gabriel et al., 2021). Therefore, it is not clear whether the divergent time-of-day effect of exercise on glycaemic regulation is due to an intrinsic metabolic disruption, or whether this phenomenon will persist in people who are metabolically healthy. Indeed, glycaemic regulation does not appear to be different in metabolically healthy participants between a single bout of morning or afternoon exercise (Tanaka et al., 2021). One must also consider the timing of various treatment strategies within the holistic diurnal environment in regards to management of type 2 diabetes. For example, metformin is the most commonly prescribed initial pharmacological intervention in type 2 diabetes and is known to alter skeletal muscle mitochondrial adaptions to exercise (Konopka et al., 2019).

Several studies suggest that metformin may interfere with the glucose-lowering effect of acute exercise (Abdalhk et al., 2020; Boulé et al., 2011; Sharoff et al., 2010). This includes an analysis of the NHANES continuous survey (1999–2018, $n = 6447$), which found that exercise and metformin do not appear to have an additive effect on glycaemia (Abdalhk et al., 2020). It is also unclear whether there is an interaction between metformin and exercise at different times of the day. Optimizing exercise timing in people with type 2 diabetes is critically important, but so is understanding any interactions with widely prescribed pharmacological treatments. In this regard, no studies have yet been able to measure time-of-day exercise effects upon diurnal blood glucose in relation to metformin timing. Furthermore, previous trials (Savikj et al., 2019; Teo et al., 2020) in people with type 2 diabetes have not assessed whether compensatory physical activity, altered diet or changed sleeping patterns underlie any time-of-day exercise effect. In this 16-week randomized crossover study, we aimed to investigate glycaemic outcomes of moderate intensity exercise at different times of the day in people with type 2 diabetes also being prescribed metformin monotherapy. Secondary outcomes included total physical activity, medication timing, diet and sleep. We wanted to find out if it is possible to optimize timing of concomitant metformin and exercise in people with type 2 diabetes.

**Brenda J. Peña Carrillo** is currently a Mexican government funded PhD Student from the University of Aberdeen. Her interest in skeletal muscle physiology began during her undergraduate studies. Now, her doctoral research aims to improve management of type 2 diabetes, focussing on exploring the interaction between metformin, exercise and skeletal muscle metabolism. Motivated by a desire to contribute to the improvement of individuals living with type 2 diabetes, Brenda's research is driven by a passionate objective to positively impact their lives through innovative scientific research. **Brendan M. Gabriel** is a principal investigator at the University of Aberdeen, and an affiliated researcher at The Karolinska Institute. His research focusses on skeletal muscle and its role in disease pathology, in addition to assessing physical activity as a treatment or preventative intervention in metabolic disease. He was part of the Diabetes UK Innovators in Diabetes (IDia) Programme (2021–2023), and was awarded the European Foundation for the Study of Diabetes (EFSD)/Lilly – Young Investigator Research Award (2020) and a Novo Nordisk Foundation Fellowship (2020–2024).

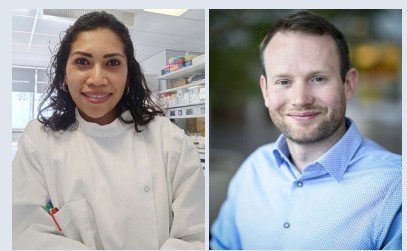

## Methods

### Ethical approval and study design

The study was conducted following ethical approval from the UK Health Research Authority's Integrated Research Application System (IRAS), and via the London Centre Research Ethics Committee (REC reference: 20/PR/0990; IRAS project ID: 292015) and in accordance with the principles of the *Declaration of Helsinki* as revised in 2008. Informed written consent was obtained from participants during recruitment. The trial was preregistered – Research Registry Unique Identifying Number: researchregistry6311. Nine males and nine females with type 2 diabetes also being prescribed metformin monotherapy completed a remote, randomized, crossover two-arm trial. An overview of the study's design is shown in Fig. 1.

### Participants

**Eligibility criteria.** To be considered eligible for inclusion, participants had to be diagnosed with type 2 diabetes (insulin independent); have a body mass index (BMI) of 20−36 kg/m$^2$; be aged 45−75 years, male or female; be prescribed metformin; be able to participate in an exercise intervention; be able to interact with smart-phone apps; and be able to provide informed consent. Participants were excluded from the study if they were treated with insulin medication; were current nicotine users (cigarettes/nicotine gum, etc.); were past nicotine users <6 months before inclusion in the study; have a pre-existing cardiovascular condition; have pre-existing blood-borne disease; have pre-existing rheumatic illness; have cancer; have a pre-existing psychiatric disorder; have another pre-existing systemic disease; have an extreme chronotype (e.g. extreme 'lark' or 'owl'); were prescribed any glycaemia regulating medications other than metformin; were already meeting physical activity guidelines, i.e. >150 min of moderate intensity activity a week or 75 min of vigorous intensity activity a week; or could not adequately read or understand English.

**Recruitment.** Volunteers were recruited from the Aberdeen and Aberdeenshire areas through the Scottish Primary Care Research Network (SPCRN), over the course of 1 year, between May 2021 and June 2022. The

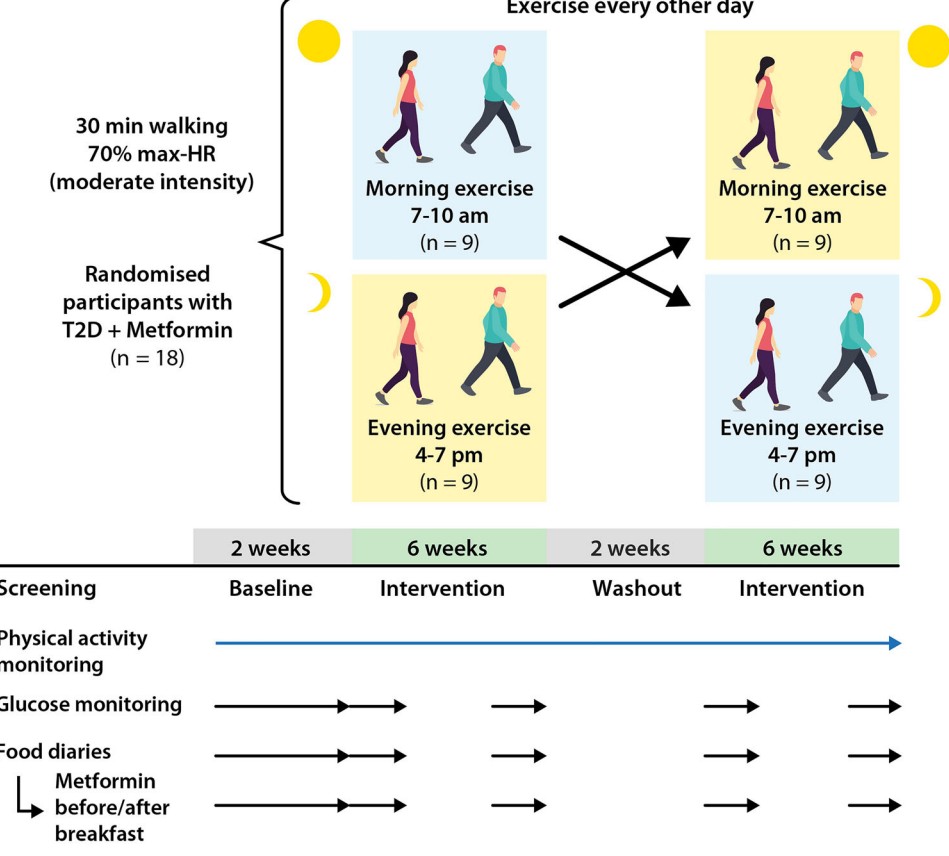

**Figure 1. Study design**
Participants engaged in randomized crossover morning or evening exercise sessions, lasting 6 weeks per arm. HR, heart rate; T2D, type 2 diabetes. Figure created with Biorender. [Colour figure can be viewed at wileyonlinelibrary.com]

experimental study included recruitment and exercise interventions for two groups, winter and summer cohorts. SPCRN conducted a pre-screening, and eligible potential volunteers received a letter of invitation from their National Health Service GP practice. If individuals were interested in participating, they were sent the consent form and questionnaires as part of an information pack. After phone consultation, participants who agreed to continue were instructed to complete, sign and to mail the forms back to the research team. Following a review of the forms to ensure they were properly completed, the necessary equipment was posted to the participants. Subsequently, participants received another phone call from a member of the research team to review the participant information sheet (PIS), answer any questions the participant may have and explain the instructions on how to fit continuous glucose monitors and physical activity monitors.

GPs were notified about their patient's participation in the study and the researchers were sent participants' information about BMI, HbA1c, date of type 2 diabetes diagnosis and time since metformin prescription from the GP records. One participant did not have any available data, five of the participants did not have available records about type 2 diabetes diagnosis date and four participants did not have available records for time since metformin prescription as they were new to the GP.

**Sample size calculation.** The number of participants (30) was determined in collaboration with an independent statistician at Biomathematics and Statistics Scotland (BioSS) based on our previous study (Savikj et al., 2019), i.e. an estimated difference in mean glucose concentration of 0.69 mmol/l, an estimated standard deviation (SD) of 1.32 mmol/l (SD of the individual differences between two exercise treatments), a paired analysis, a significance level of 0.05 and a power of 80%.

**Randomization to morning or evening exercise.** Randomization was carried out by an independent statistician at BioSS. Microsoft Excel was used to randomly allocate participants to the two possible sequences (i.e. morning first or evening first) in blocks by order of joining the trial and sex. The randomization was generated once we received a positive response from the participant.

**Adherence monitoring.** Compliance with the exercise regimen were assessed using real-time data obtained from the Garmin Connect (Garmin Ltd, Olathe, KS, USA) and LibreView (Abbott Diabetes Care Inc., Alameda, CA, USA) sites. Participants missing pre-defined adherence targets (i.e. either missing >4 exercise windows in less than 2 weeks, with a heart rate discrepancy of >15%

during exercise or >4 days with no glucose reading) were contacted by phone call or e-mail and encouraged to improve adherence. If adherence was not improved, they were not considered for the final data analysis. When the trial coincided with the Christmas holidays, the wash-out period was extended for a period of 4 weeks so that the intervention arms did not run during the Christmas holidays.

## Exercise intervention

Participants were requested to perform 30 min of walking every other day during a 6-week period, maintaining, as close as possible, 70% of their estimated maximum heart rate (estimated with the formula 220 minus age; Fox et al., 1972) during exercise. Participants were instructed to perform exercise within a 3-h window between either 07.00 and 10.00 h or 16.00 and 19.00 h. The overall 16-week trial period included a 2-week baseline recording, 12-week intervention arms (6 weeks morning and then 6 weeks evening exercise, or vice versa) with a 2-week washout between trial arms. Physical activity data were collected for the entirety of the trial, pre-trial and wash-out period.

## Outcomes

**Physical activity.** Physical activity data (assessed as step count per day and heart rate) as well as sleep quality were collected via a wrist-based physical activity monitor, Garmin Vivosmart 4 (Garmin Ltd), this device can accurately measure physical activity variables with a low percentage of error: heart rate (HR), 5.56 bpm; steps-per-day, 36.75 (Terasawa et al., 2023). This device can also detect changes in sleep variables, especially during long-term monitoring (Mouritzen et al., 2020). Participants were asked to wear the physical activity monitors for the entire duration of the trial and were instructed to download the Garmin Connect application and log in with a pre-determined study password. The outcomes of the physical activity monitors were step count per day, heart rate and sleep quality. The participants' chronotype, reflecting individual preferences in sleep–wake rhythms, was identified through the self-administered Munich Chronotype Questionnaire, consisting of 29 questions; the scale assesses wake and sleep patterns on both work and free days, energy levels throughout the day, sleep latency and exposure to daylight.

**Glucose monitoring.** Participants were provided with continuous glucose monitors, The FreeStyle Libre 2 sensor (CGMs, Abbott Diabetes Care Inc., Alameda, CA, US), that measures the glucose concentration in interstitial fluid continuously, storing a reading every 15 min. Sensors

had a working life of 2-weeks. Glycaemia data was collected by the LibreView (Abbott Diabetes Care Inc.) cloud system; participants were instructed to download the FreeStyle LibreLink application on their own smart phones to share and link their data to a LibreView account in real time. Glucose information was collected for 2 weeks before the start of the intervention, during the first 2 weeks, and for the last 2 weeks of each exercise intervention.

**Dietary intake and metformin dose.** Participants were asked to maintain their normal diet consistently throughout the trial and record their diet throughout the trial using a 4-day food diary in which the participant recorded the food eaten, the time of day each meal was consumed, the cooking methods used and the portion sizes. A 4-day food diary was completed during the baseline period, and during the first 2 weeks, and the last 2 weeks of each exercise intervention. Dietary intake was determined using Windiet software (Univation Ltd, The Robert Gordon University, Aberdeen, UK). During the analysis, we divided the dietary intake into breakfast, lunch and dinner for comparison of time or size of meal intake. Snacks were combined with the nearest meal by time. Participants were also asked to record the time and dose of metformin on their food diaries. As dietary intake and metformin time and dose were self-reported, the data analysis of the food diaries and the metformin analysis before or after breakfast corresponded to 14 participants that fulfilled this requirement (for baseline period, metformin pre-breakfast ($n = 9$), metformin post-breakfast ($n = 5$); morning exercise, metformin pre-breakfast ($n = 8$), metformin post-breakfast ($n = 5$); evening exercise, metformin pre-breakfast ($n = 10$), metformin post-breakfast ($n = 4$)). Metformin analysis before or after dinner corresponded to 13 participants (for baseline period, metformin pre-dinner ($n = 10$), metformin post-dinner ($n = 3$); morning exercise, metformin pre-dinner ($n = 9$), metformin post-dinner ($n = 4$); evening exercise, metformin pre-dinner ($n = 7$), metformin post-dinner ($n = 6$)).

### Data analysis

Blood glucose levels were measured using CGMs, from which 24-h hourly means were calculated using the RStudio statistical software package v2022.12 (R version 4.2.2; RStudio, PBC, Boston, MA, USA). Data was then analysed using Prism 9.5.1 software (GraphPad Software, Boston, MA, USA). Adherence to exercise protocol variables were compared by means using Student's paired *t* test. Baseline characteristics between males and females, winter and summer, metformin before/after breakfast

**Table 1. Baseline characteristics of study participants**

| Characteristic ($n = 18$) | Mean $\pm$ SD |
|---|---|
| Sex | Male $n = 9$; female $n = 9$ |
| Age (years) | 61 $\pm$ 8.2 |
| BMI (kg/m$^2$) | 30.6 $\pm$ 3.0 |
| HbA1c (mmol/mol) | 64.4 $\pm$ 15.0 |
| HbA1c (%) | 8.0 $\pm$ 1.4 |
| Time since T2D diagnosed (years) | 8.2 $\pm$ 6.4 |
| Dose of metformin per day (mg) | 1313.0 $\pm$ 466.2 |
| Time of metformin (years) | 5.2 $\pm$ 3.5 |
| Steps per day at baseline | 6798 $\pm$ 2523 |

T2D, type 2 diabetes.

were compared by means using an unpaired *t* test. Changes in glucose during exercise intervention and caloric intake were analysed by one-way or two-way ANOVA followed up by the Holm–Šídák multiple comparisons test. Given that participants recorded time and doses of metformin intake, the areas under the curve (AUC) for glucose level were categorized according to whether metformin was taken before or after breakfast and dinner, and these categories were analysed by an unpaired *t* test. Significance level was set at $P < 0.05$. Results are expressed as means $\pm$ SD.

### Results

#### Participants recruitment and characteristics

Of the 40 positive responses received, nine males and nine females with type 2 diabetes being prescribed metformin completed the 16-week crossover intervention; see Fig. 2 for Consolidated Standards of Reporting Trials (CONSORT) flow chart. Following randomization, eight participants withdrew for personal reasons, four of them from the winter cohort and four from the summer group. After completion of the trial, five participants did not meet the exercise intervention requirements, so at the finish there were 11 participants from the winter and seven from the summer cohort. In terms of crossover sequence, the study finished with eight participants who did morning exercise first and 10 participants who did evening exercise first. Sleep data correspond to 15 participants who fulfilled this requirement; and glucose data were missing from one participant during the last 2 weeks of the morning period and from two participants for the last 2 weeks of evening exercise. Baseline characteristics of study participants that completed the trial are shown in Table 1 (Characteristics according to different stratification are shown in Supporting information, Tables S1–S3).

## Adherence to exercise intervention

The total number of exercise sessions participants were asked to complete during each exercise period was 21. The exercise completion rate was consistent ($P = 0.47$) between morning ($19.1 \pm 5.5$ sessions) and evening ($18.1 \pm 5.8$ sessions) exercise (Supporting information, Fig. S1). On average, participants completed sessions at $08.46 \pm 00.54$ h in the morning and $16.48 \pm 00.40$ h in the evening (Supporting information, Fig. S2). Total physical activity was similar between the arms of the trial (Fig. 3*A* and *B*). The number of mean daily steps recorded during baseline was $6798.0 \pm 2523$. During

exercise days participants completed a mean daily total of $10,814.0 \pm 2251$ and $10,373.0 \pm 2183$ steps during morning and evening sessions, respectively ($P = 0.15$). During rest days participants completed a mean daily total of $6843.0 \pm 2383$ and $6344.0 \pm 2182$ steps during morning and evening sessions, respectively ($P = 0.24$). The total hours of sleep registered from Garmin devices ($8.3 \pm 1.1$) was significantly higher ($P = 0.03$) than the hours of sleep recorded from self-reported MCTQ ($7.67 \pm 0.9$) (Supporting information, Fig. S3). The total hours of sleep indicated a trend (one-way ANOVA: $P = 0.06$) showing longer sleep duration during evening exercise ($8.3 \pm 1.1$ h) compared with morning exercise

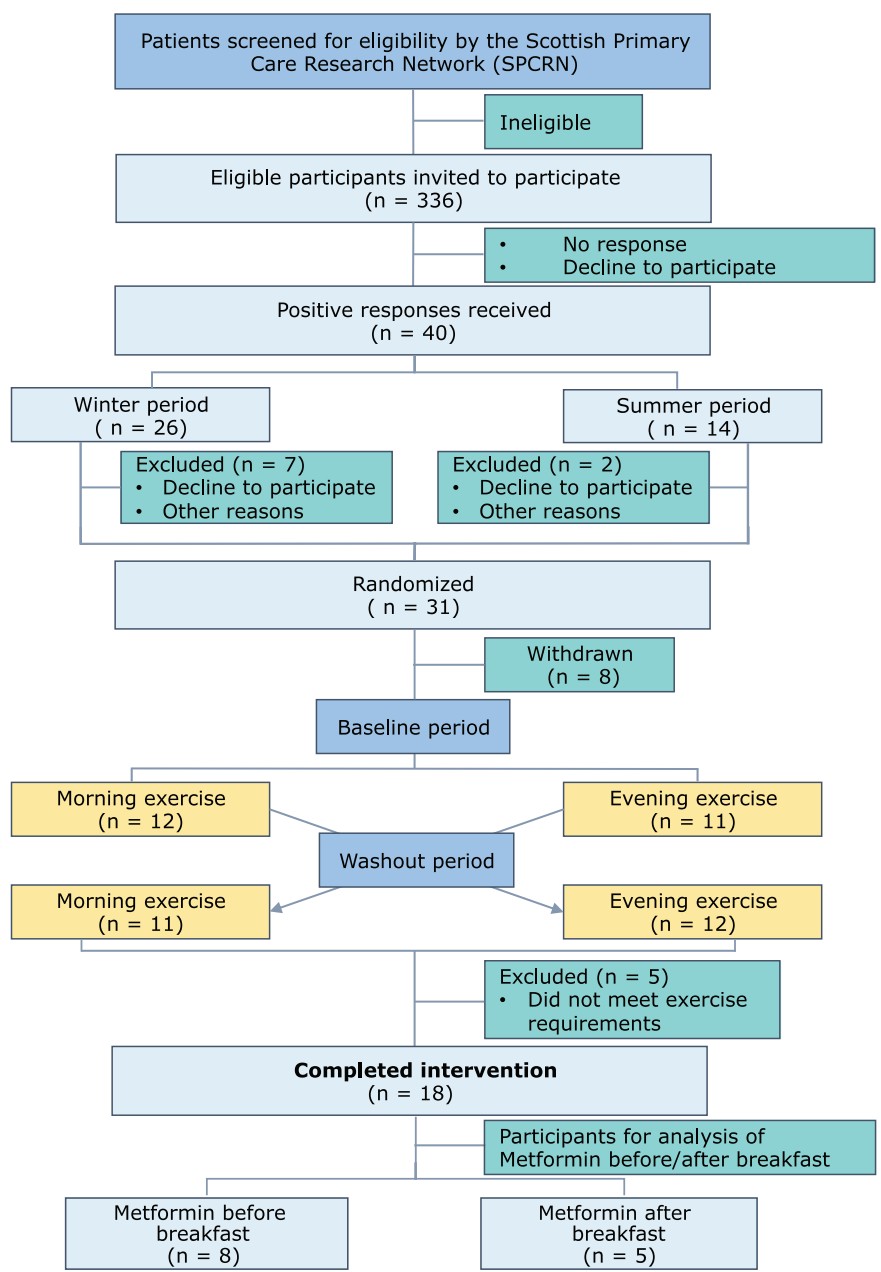

**Figure 2. CONSORT Flow Chart**
Consolidated Standards of Reporting Trials (CONSORT) flow chart. [Colour figure can be viewed at wileyonlinelibrary.com]

(7.9 ± 0.3 h); however there was no difference compared with baseline ($P = 0.31$ and $P = 0.40$ for morning and evening exercise, respectively) (Fig. 3C). Sleep architecture such as deep sleep, light sleep, REM sleep and time awake showed no significant difference between baseline, morning and evening exercise ($P = 0.35$, $P = 0.57$, $P = 0.06$, $P = 0.20$, respectively) (Supporting information, Fig. S4). Participants were instructed to exercise at an estimated 70% of max-HR, which was 111.4 ± 5.5 bpm. Exercise intensity during morning and evening exercise was similar (Fig. 3D), as measured by HR (117.2 ± 8.2 and 117.3 ± 11.5 bpm, respectively, $P = 0.93$). Overall, our results show no differences in physical activity or compensatory activity between morning and evening exercise.

## Changes in glucose during exercise intervention

There was a significant decrease ($P = 0.02$) in acute (first 24 h of exercise) glucose AUC during morning exercise (180.6 ± 68.4 mmol/l) compared with baseline (210.3 ± 76.7 mmol/l), whereas there were no significant differences ($P = 0.12$) in acute glycaemia levels during evening exercise (191.7 ± 48.53 mmol/l) compared with baseline (210.3 ± 76.7 mmol/l) (Fig. 4A). No change in acute glucose levels between morning and evening exercise were observed ($P = 0.14$) (Fig. 4A). An

hour-by-hour time course of glucose readings did not indicate a difference in glycaemia during the trial ($P > 0.05$) (Fig. 4B).

Twenty-four-hour AUC for glucose during baseline (210.3 ± 76.7 mmol/l), morning (197.3 ± 58.96 mmol/l) and evening (206.7 ± 63.22 mmol/l) during weeks 1−2 did not differ significantly ($P = 0.24$) (Fig. 5A). Further, no difference in glucose concentration was observed at any time point during the 24-h time course between baseline, morning and evening exercise, during weeks 1−2 (Fig. 5B). Similar findings were observed at weeks 5−6 (Fig. 5D).

No differences in physiological characteristics were observed between males and females (Table 2). Furthermore, no significant interaction effect in glucose AUC was observed for sex × time of the exercise (Supporting information, Fig. S5).

Physiological characteristics of participants showed lower HbA1c levels ($P = 0.04$) during summer (55.6 ± 9.3 mmol/mol) compared to winter (70.7 ± 15.5 mmol/mol) (Table 3). There was a trend ($P = 0.07$) showing lower glucose AUC levels in summer (169.8 ± 2.1 mmol/l) compared to the winter (222.4 ± 14.3 mmol/l). This consistent pattern persisted throughout the entire intervention period; however, this difference in glucose AUC levels between winter and summer did not reach statistical significance ($P = 0.07$) (Supporting information, Fig. S6).

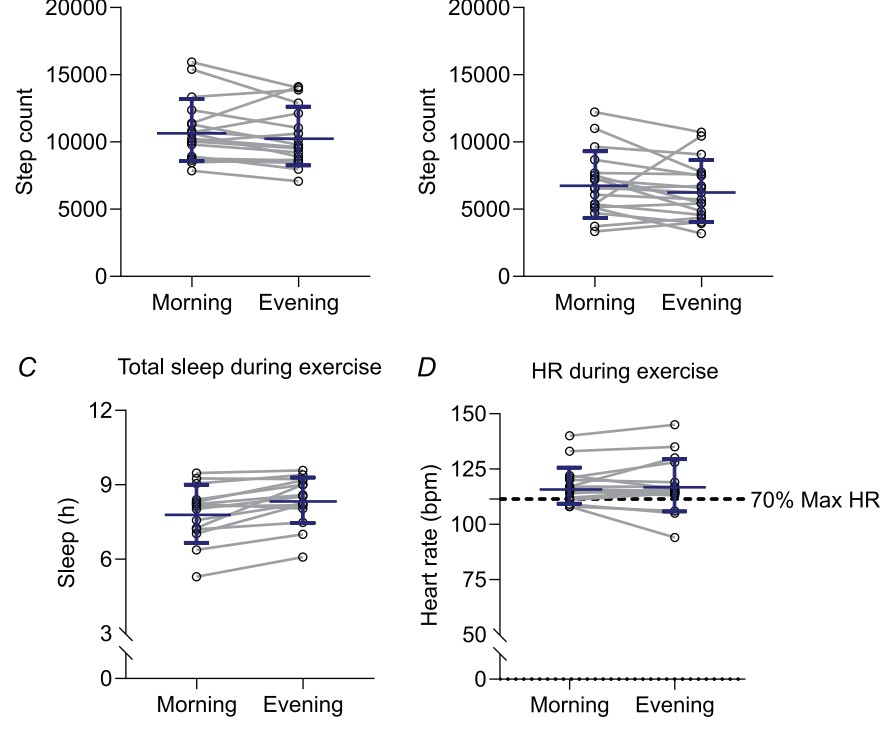

**Figure 3. Step count, max-HR and total sleep hours during the morning and evening exercise periods**
*A* and *B*, number of steps on days when the participants performed exercise (*n* = 18) (*A*) and rest days (*n* = 18) (*B*). *C* and *D*, total sleep hours during exercise (*n* = 15) (*C*) and Max-HR (*n* = 18) throughout trial (*D*). Blue lines are means ± SD, dashed line (*D*) represents 70% Max-HR estimated. HR, heart rate. Step count, total hours of sleep and sleep insights were analysed using one-way ANOVA; total hours of sleep from Garmin devices *vs*. MCTQ and Max-HR were analysed using a paired *t* test. [Colour figure can be viewed at wileyonlinelibrary.com]

**Table 2. Physiological characteristics by gender**

| Characteristic | Males (*n*) | Females (*n*) | *P* |
|---|---|---|---|
| Age (years) | 58.3 ± 8.3 (9) | 63.6 ± 7.5 (9) | 0.18 |
| BMI (kg/m$^2$) | 31.4 ± 3.0 (9) | 29.8 ± 3.1 (8) | 0.28 |
| HbA1c (mmol/mol) | 65.7 ± 16.8 (9) | 63.1 ± 13.9 (8) | 0.74 |
| HbA1c (%) | 8.1 ± 1.5 (9) | 7.9 ± 1.3 (8) | 0.71 |
| Time since T2D diagnosed (years) | 4.0 ± 2.2 (5) | 10.6 ± 6.9 (9) | 0.06 |
| Dose of metformin per day (mg) | 1500.0 ± 500.0(3) | 1250 ± 467.7(9) | 0.45 |
| Time of metformin (years) | 3.6 ± 2.1 (5) | 6.2 ± 4.0 (8) | 0.20 |
| Number of steps | 7114 ± 2953 (9) | 6481 ± 2140 (9) | 0.60 |

HbA1c, blood glycosylated haemoglobin. Values are mean ± SD (*n*). *P*-values are from unpaired *t* test.

**Table 3. Physiological characteristics by season**

| Characteristic | Winter (*n*) | Summer (*n*) | *P*-value |
|---|---|---|---|
| Age (years) | 60.8 ± 7.1 (11) | 61.3 ± 10.3 (7) | 0.90 |
| BMI (kg/m$^2$) | 30.4 ± 2.9 (10) | 31.0 ± 3.4 (7) | 0.69 |
| HbA1c (mmol/mol) | **70.7 ± 15.5 (10)** | **55.6 ± 9.3 (7)** | **0.04** |
| HbA1c (%) | **8.7 ± 1.3 (10)** | **7.3 ± 0.8 (7)** | **0.03** |
| Time since T2D diagnosed (years) | 9.4 ± 8.3 (7) | 7.0 ± 4.2 (7) | 0.50 |
| Dose of Metformin per day (mg) | **1042.0 ± 245.8 (5)** | **1583.0 ± 491.6 (6)** | **0.04** |
| Time of Metformin (years) | 4.5 ± 4.3 (6) | 5.8 ± 2.9 (7) | 0.51 |
| Number of steps | 6184 ± 2175 (11) | 7761 ± 2893 (7) | 0.20 |

HbA1c, blood glycosylated haemoglobin. Values are means ± SD (*n*). *P*-values are from unpaired *t* test. Bold values signify *P* < 0.05.

## Caloric intake during exercise intervention

The breakfast time was later (*P* = 0.02) during the evening arm (09.17 ± 00.52 h) compared to the baseline period (08.57 ± 00.49 h); however, there were no differences (*P* = 0.30) between baseline (08.57 ± 00.49 h) and the morning arm (09.04 ± 00.38 h). No significant differences (*P* = 0.49) in dinner times were observed during baseline (18.34 ± 00.32 h), morning (18.31 ± 00.40 h) or evening exercise (18.40 ± 00.53 h) (Supporting information, Fig.

S7). There were no changes in dietary energy intake during breakfast (*P* = 0.30), baseline (390.5 ± 142.9 kcal), morning weeks 1−2 (355.0 ± 85.24 kcal), evening weeks 1−2 (353.1 ± 192.3 kcal), morning weeks 5−6 (311.6 ± 101.8 kcal), or evening weeks 5−6 (354.2 ± 146.8 kcal) throughout the trial (Fig. 6). Energy intake during lunch was significantly different (*P* = 0.03) between morning weeks 1−2 (534.3 ± 228.4 kcal) and evening weeks 5−6 (402.7 ± 164 kcal) (Fig. 6). However,

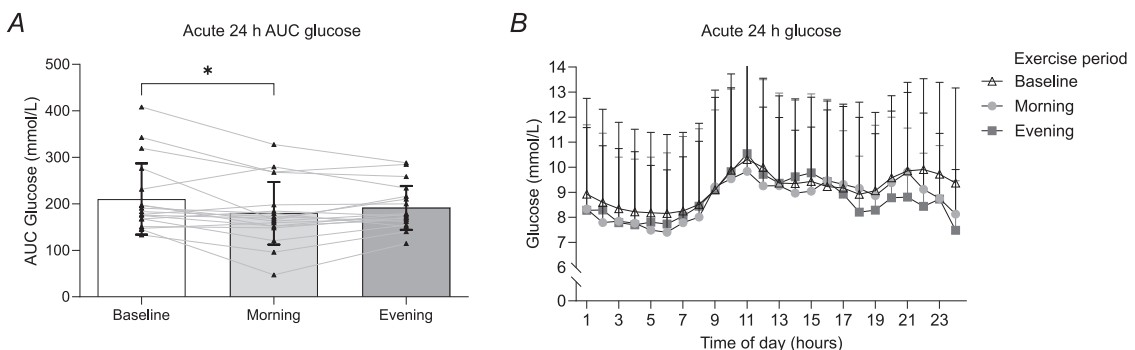

**Figure 4. Acute 24-h glucose levels**
*A*, area under the curve (AUC) values for glucose levels during the first 24 h of exercise. (*n* = 18). *B*, time course of hourly glucose levels for the first 24 h of exercise during baseline, morning and evening intervention (*n* = 18). Wk, week. **P* < 0.05. Values are means ± SD, and lines represent individual values. Data were analysed using one-way ANOVA followed up by the Holm–Šídák multiple comparisons test.

no significant differences were observed ($P = 0.64$) when comparing evening weeks 5−6 with the energy intake at baseline (470.4 ± 154.6 kcal). There was no difference in caloric ingestion of dinner ($P = 0.14$) (Fig. 6) between arms of the trial. No changes in caloric intake were detected between comparable periods of the trial.

There were no differences ($P = 0.33$) in total daily caloric intake consumed during baseline (570.6 ± 246.1 kcal), morning weeks 1−2 (573.3 ± 259.6 kcal), evening weeks 1−2 (533.1 ± 202.3 kcal), morning weeks 5−6 (585.2 ± 286.9 kcal) and evening weeks 5−6 (545.2 ± 289.8 kcal) (Fig. 7). Throughout the trial, caloric intake was significantly higher ($P \leq 0.001$) during dinner (843.1 ± 53.1 kcal), compared with breakfast (349.0 ± 28.9 kcal) and lunch (492.4 ± 61.0 kcal).

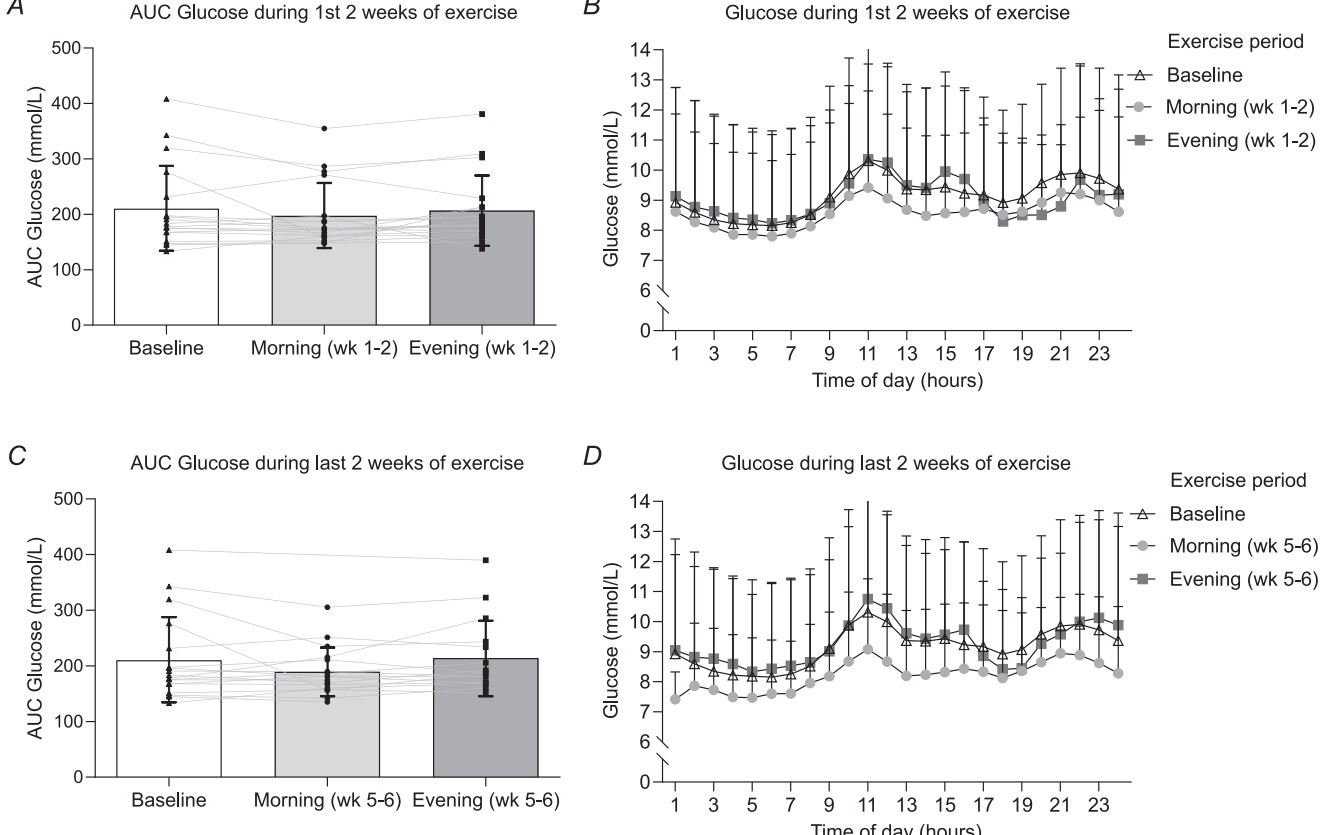

**Figure 5. Glucose levels in response to exercise intervention**
*A*, area under the curve (AUC) for 24-h glucose for week 1−2 of exercise intervention (*n* = 18). *B*, time-course plot of 24-h glucose in weeks 1−2 (*n* = 18). *C*, area under the curve (AUC) for 24-h glucose for week 5−6 of exercise intervention (*n* = 17). *D*, time course plot for 24-h glucose in weeks 5−6 (*n* = 17). Wk, week. Values are means ± SD, lines represent individual values. Data were analysed using one-way ANOVA followed up by the Holm–Šídák multiple comparisons test.

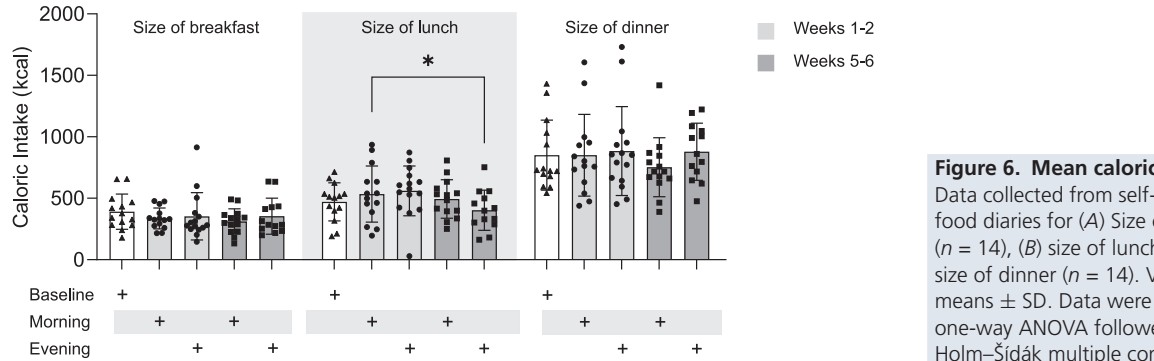

**Figure 6. Mean caloric intake**
Data collected from self-reported 4-days food diaries for (*A*) Size of breakfast (*n* = 14), (*B*) size of lunch (*n* = 14) and (*C*) size of dinner (*n* = 14). Values are means ± SD. Data were analysed using one-way ANOVA followed up by the Holm–Šídák multiple comparisons test.

## Meals and metformin timing

Only when the participants performed morning exercise was acute glucose AUC lower ($P = 0.01$) in participants taking metformin before breakfast ($152.5 \pm 29.95$ mmol/l) compared with participants taking metformin after breakfast ($227.2 \pm 61.51$ mmol/l) (Fig. 8). No differences were observed for metformin before or after breakfast for the baseline and evening exercise arms of the trial ($P = 0.17$ and $P = 0.67$, respectively), nor for metformin before or after dinner during baseline or morning or afternoon exercise ($P = 0.44$ and $P = 0.25$, respectively) (Supporting information, Fig. S8).

During morning exercise (weeks 1−2), participants taking metformin before breakfast exhibited a trend ($P = 0.07$) to have lower glucose AUC levels ($165.4 \pm 9.3$ mmol/l) compared with participants taking metformin after breakfast ($223.3 \pm 61.4$ mmol/l) (Fig. 9). During weeks 5−6 of the exercise protocol, glucose was significantly lower ($P = 0.04$) for participants taking metformin before breakfast ($168.8 \pm 15.8$ mmol/l), rather than after breakfast ($224.5 \pm 52.0$ mmol/l) only during morning exercise. No differences in glucose AUC for metformin before/after breakfast were observed for the evening arm of the trial throughout the protocol (evening arms, weeks 1−2, $P = 0.80$, weeks 5−6 $P = 0.72$).

In the morning exercise period, a glucose difference ($P = 0.03$) was observed at 22.00 h between metformin before and after breakfast (Fig. 10*B*). No differences were observed at any time point during the 24-h time course between metformin before and after breakfast for baseline and evening exercise (Fig. 10*A* and *C*).

## Discussion

These data demonstrate that morning moderate intensity exercise acutely reduces glycaemia in people with type 2 diabetes also being prescribed metformin, while evening

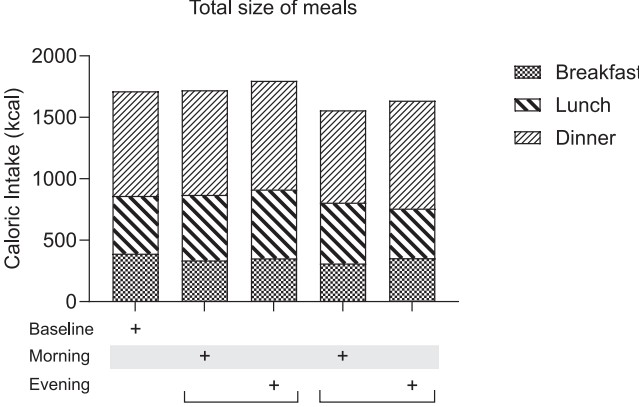

**Figure 7. Total daily caloric intake**
Data collected from self-reported 4-day food diaries from baseline ($n = 14$), morning weeks 1−2 ($n = 14$), evening weeks 1−2 ($n = 15$), morning weeks 5−6 ($n = 14$) and evening weeks 5−6 ($n = 13$). Values are means ± SD. Data were analysed using one-way ANOVA followed up by the Holm–Šídák multiple comparisons test.

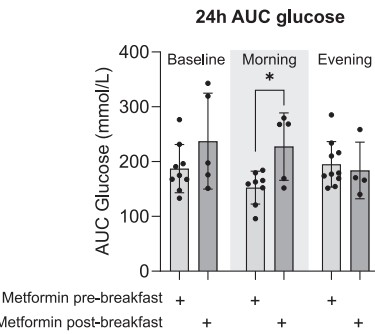

**Figure 8. Acute Glucose AUC in response to Meal and metformin timing**
Area under the curve (AUC) values for glucose levels during the first 24 h of exercise for baseline, metformin pre-breakfast ($n = 9$), metformin post-breakfast ($n = 5$) (*A*), morning exercise, metformin pre-breakfast ($n = 8$), metformin post-breakfast ($n = 5$) (*B*) and evening exercise, metformin pre-breakfast ($n = 10$), metformin post-breakfast ($n = 4$) (*C*). *$P < 0.05$. Values are means ± SD. Data were analysed using unpaired *t* test.

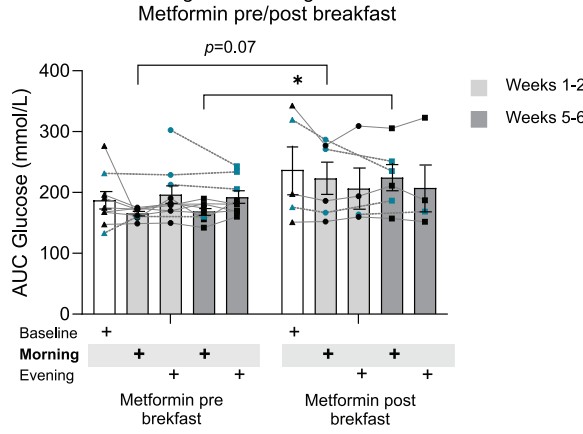

**Figure 9. Glucose AUC values throughout the trial**
Area under the curve (AUC) values for glucose levels for Baseline period, metformin pre-breakfast ($n = 9$), metformin post-breakfast ($n = 5$); morning exercise (weeks 1−2), metformin pre-breakfast ($n = 8$), metformin post-breakfast ($n = 6$); evening exercise (weeks 1−2), metformin pre-breakfast ($n = 10$), metformin post-breakfast ($n = 4$); morning exercise (weeks 5−6), metformin pre-breakfast ($n = 8$), metformin post-breakfast ($n = 6$); and evening exercise (weeks 5−6), metformin pre-breakfast ($n = 9$), metformin post-breakfast ($n = 4$). *$P < 0.05$. Values are means ± SD. Blue data points are participants that changed metformin intake timing between different periods of the trial, and lines represent individual crossover values. Data were analysed using two-way mixed-model ANOVA followed up by the Holm–Šídák multiple comparisons test. [Colour figure can be viewed at wileyonlinelibrary.com]

exercise had no effect upon glycaemia. The acute reduction of glycaemia in the morning trial was apparently driven by people who consumed metformin before breakfast, rather than after breakfast. Indeed, morning exercise combined with pre-breakfast metformin persistently reduced glucose AUC compared to morning exercise combined with post-breakfast metformin up to the final week (week 6) of the intervention. In this crossover trial, there was no significant difference in exercise intensity, caloric intake or total physical activity between the arms of the trial (morning *vs.* evening exercise). These findings support the assertion that an interaction with intrinsic circadian biology may be linked to the observed effects.

Clinicians often recommend taking metformin with or after meals, including breakfast (e.g. National Institute for Health and Care Excellence guidelines, November 2023). However, many people taking metformin struggle to maintain regular physical activity (Krug et al., 1991), which may be a result of a lower tolerance of exercise when metformin is concomitant (Kristensen et al., 2019). Furthermore, metformin may interfere with the glucose-lowering effect of acute exercise (Boulé et al., 2011; Sharoff et al., 2010). Therefore, identifying strategies to optimize concomitant prescription of these therapies is essential. Our data suggest that it may be possible to optimize timing of these concomitant therapies to augment their therapeutic effect. It should be noted that the specific data in the current study are not crossover; however, the magnitude of difference (33% decreased glucose AUC in pre-breakfast group, Fig. 8) suggests a real effect. Additionally, an interaction effect on glycaemia between morning exercise and pre-breakfast metformin intake appeared to persist throughout the trial since the group taking metformin before breakfast had lower glucose AUC than the post-breakfast group into week 6 of the morning exercise intervention. These findings have parallels to previous data that suggest consuming metformin before a meal improved efficacy upon glycaemic regulation (Hashimoto et al., 2016). This effect on glycaemia may be partly related to metformin's

pharmacokinetic interaction with meal intake and intrinsic circadian rhythms. For example, compared with the fasting state, bioavailability of metformin is 24% lower, and the peak concentration delayed about 37 min when an 850 mg tablet is administered with food (Sambol et al., 1996). Additionally, metformin's pharmacology significantly depends on time-of-day in humans, which may be related to glomerular filtration rate, renal plasma flow and renal organic cation transporter (OCT) 2 activity (Türk et al., 2023). Although the half-life of metformin in the blood is relatively long ($\sim$18 h depending on dose/method; Larsen et al., 2012), less is known regarding accumulation in skeletal muscle, a key organ in the response to exercise (Gabriel & Zierath, 2017). It is known that concomitant exercise alters the pharmacokinetics of acute metformin administration (Kristensen et al., 2019; Larsen et al., 2012), with differing skeletal muscle concentrations of metformin before, during and after exercise (Kristensen et al., 2019). It is plausible then, that exercise at different times of the day in addition to timing of metformin intake may interact with the pharmacokinetics and glycaemic modulating effects of metformin. This is supported by our findings since we observed a glucose lowering effect of pre-breakfast metformin only during morning exercise periods of the trial. Thus, further research should aim to fully elucidate the interaction between meal-timing, exercise and medication intake.

Another possible diurnal phenomenon that may contribute to our observed effects is that of the potentiation of pre-breakfast metformin upon the effect of fasted morning exercise upon blood glucose. The effect of fed *vs.* fasted exercise on glycaemic control has been debated in the literature with differing outcomes dependent on experimental conditions and the populations studied (Wallis & Gonzalez, 2019). In our study, we did not specify that participants should perform exercise before or after breakfast, and the mean time of morning exercise was 08.46 $\pm$ 00.54 h, while breakfast was at 09.04 $\pm$ 00.38 h during the morning

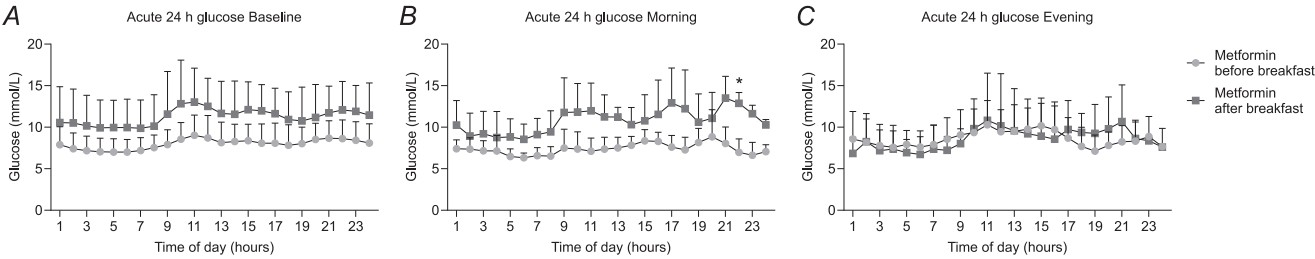

**Figure 10. Time course of acute glucose levels in response to metformin timing**
Time course of 24-h glucose levels in response to exercise intervention Baseline, metformin pre-breakfast ($n = 9$), metformin post-breakfast ($n = 5$) (*A*), morning exercise, metformin pre-breakfast ($n = 8$), metformin post-breakfast ($n = 5$) (*B*), and evening exercise, metformin pre-breakfast ($n = 10$), metformin post-breakfast ($n = 4$) (*C*). *$P < 0.05$. Values are means $\pm$ SD. Data were analysed using two-way ANOVA followed up by the Holm–Šídák multiple comparisons test.

exercise period. Therefore, our data suggest it is possible that pre-breakfast metformin does have a potentiation effect on fasted morning exercise.

In this study we recruited two cohorts in the summer and winter to account for seasonal differences in our observations. We were thus able to stratify participants into winter and summer cohorts; however, the study was not designed to detect primary outcomes in this manner. There were no statistically significant differences between winter and summer cohorts, although the winter cohort did have consistently higher glucose AUC levels in all periods of the trial. An observational study performed in a Canadian healthcare setting found that hospital encounters for hyperglycaemia peaked in January compared with the other calendar months (11% increase) (Clemens et al., 2017). Although seasonal differences in glycaemic regulation are clearly multifactorial, optimization of management strategies for type 2 diabetes should also consider seasonal variations.

In addition to seasonal differences, long-term acclimation to the exercise intervention may be an important factor to consider. We observed significantly lower glucose levels during the morning exercise period in people who took metformin before breakfast compared to after breakfast both acutely (initial 24 h) and in the final 2 weeks of the morning exercise period. However, we observed only a trend ($P = 0.07$) for lower glucose in the first 2 weeks of exercise when participants were stratified in the same manner. It is unclear whether this is due to statistical power, or whether there is an acute and training acclimation effect in regards to the timing interaction between exercise and metformin intake. There have been few studies assessing divergent diurnal outcomes of exercise in people with type 2 diabetes over a longer training period. Retrospective studies suggest that even over a longer exercise training period, timing of exercise may be important in glycaemic management (Qian et al., 2023); however, these findings should also be tested in specifically designed longitudinal studies.

As described above, skeletal muscle plays a key role in the metabolic response to exercise, and this organ has an intrinsically disrupted rhythm in people with type 2 diabetes (Gabriel et al., 2021). However, it is plausible that metformin treatment may interact with skeletal muscle circadian metabolism to alter time-of-day exercise effects observed in other studies in people with impaired metabolism. Supportive of this is the lack of effect upon glycaemia in response to exercise at different times of the day in people who are metabolically healthy (Tanaka et al., 2021) and who presumably do not have disrupted metabolic circadian rhythms. However, the aforementioned study (Tanaka et al., 2021) only monitored response to a single bout of exercise, while the current study was a 16-week trial in which participants did two 6-week training periods. Moderate intensity exercise did not increase glycaemia in people with type 2 diabetes being prescribed metformin. These data indicate that in terms of glycaemic response, morning moderate intensity exercise is not deleterious in people with type 2 diabetes being prescribed metformin and does not exacerbate the dawn phenomenon in the same manner as high-intensity interval training (HIIT) (Savikj et al., 2019). Indeed, exercise modality may interact with circadian rhythm to alter exercise outcomes. High-intensity exercise capacity is consistently higher in the afternoon/evening compared to morning, while diurnal differences in exercise capacity are less clear with moderate intensity exercise (Gabriel & Zierath, 2019). This phenomenon may underlie the findings of Teo et al. (2020), who found no time-of-day differences in glycaemic outcomes in people with type 2 diabetes in response to combined walking exercise and resistance training (Teo et al., 2020). This exercise modality (Teo et al., 2020) contrasts with the HIIT protocol used in previous studies with conflicting findings (Savikj et al., 2019). However, it should be noted that Teo et al. (2020) did not measure glycaemic outcomes in a circadian manner (Teo et al., 2020), whereas in the current study we have measured 24-h glucose over a total period of 10 weeks. Additionally, participants completed moderate intensity exercise outdoors in Scotland, which may influence diurnal outcomes compared to exercise protocols completed at a higher intensity or in a different environment. In a laboratory setting, 50 min of walking at three different times of day and at different timing in relation to meals did not lower 24-h glucose concentrations in people with type 2 diabetes (Munan et al., 2020). Together with the results from the current study, these data suggest that moderate intensity exercise may not give rise to time-of-day-dependent glycaemic perturbations of the same magnitude as high intensity exercise.

The current study had some limitations. Our prior statistical power calculations suggested a sample size of 30 in order to detect differences in hour-by-hour glucose readings. Although we met our recruitment target, we had a higher-than-expected withdrawal rate, with 18 participants eligible for final analysis. We conducted this study mainly during the COVID-19 global pandemic, which directly and indirectly impacted on participants' ability to complete the 16-week trial. Nevertheless, the strength of our study is the crossover design, giving greater statistical power than a cross-sectional design. We aimed to recruit people with type 2 diabetes who were being prescribed metformin monotherapy, i.e. no other glycaemic regulating pharmaceutical treatments. This criterion allowed us to investigate a time-of-day exercise interaction with metformin treatment specifically, rather than being confounded by other pharmaceutical treatments. As previously noted, metformin intake timing was not a primary outcome

of this study, and therefore our comparisons between different metformin intake timings are not crossover. Thus, it is possible that existing differences in glycaemic regulation between groups are responsible for some of the observed differences (e.g. Supporting information, Table S2). However, it is likely that metformin intake timing is highly habitual, and it is plausible that participants who habitually take metformin before breakfast may experience better long-term control of glycaemia as a direct result. A further strength of the study was the excellent adherence of participants to the exercise protocol, with >85%–90% exercise session completion rate over 6 weeks and no difference in self-monitored exercise intensity between arms of the trial. The advantage of conducting self-monitored exercise trials is that this is likely to better recapitulate many real-world settings in which clinicians' recommendations to exercise are mostly self-monitored rather than monitored by practitioners. Participants were instructed to maintain their normal diet during the trial, and total caloric intake was not different between arms of the trial, indicating adherence to this instruction. However, it should be noted that dietary intake was self-reported. We also note that we used Garmin Vivosmart 4 to monitor sleep duration and quality. As noted above, this method may not be as precise for describing sleep architecture as gold-standard techniques such as polysomnography.

We identified a discrepancy between the self-reported MCTQ and the sleep duration reported by Garmin Vivosmart 4. It is known that retrospective self-reported sleep duration can differ substantially from actigraphy and polysomnography assessments (Matthews et al., 2018). Therefore, this is a likely cause of this discrepancy. Our findings show a trend for sleep duration to be longer during evening exercise periods compared with morning exercise, although neither differed from baseline. While evening exercise has been shown to reduce sleep quality (Saidi et al., 2023) in certain populations, this is not consistent across the literature (Frimpong et al., 2021; Myllymaki et al., 2011; Youngstedt et al., 1999), and may be dependent on chronotype (Saidi et al., 2023). Thus, it is not clear if our finding is due to a lack of statistical power, or whether the null-finding is true. Additionally, evening exercise may impact differently on the sleep of those with an earlier chronotype.

In summary, our study demonstrates an acute decrease in glycaemia in response to morning exercise in people with type 2 diabetes also being prescribed metformin. The reduction of glycaemia was driven by participants who consumed metformin before breakfast, rather than after breakfast, indicating an interaction between meal-timing, metformin intake and exercise. In this trial we monitored diet, exercise intensity, and physical activity throughout the trial, none of which appeared to contribute to the time-of-day-dependent effect of exercise on glycaemia. Many people being prescribed metformin struggle to maintain regular physical activity (Krug et al., 1991), and metformin may interfere with the glucose-lowering effect of acute exercise (Boulé et al., 2011; Sharoff et al., 2010). Thus, finding strategies to augment the therapeutic effect of these treatments when concomitant is essential in the management of type 2 diabetes. Our research indicates that it may be possible to optimize the timing recommendations for concomitant exercise and metformin treatment. Specifically, our findings suggest that morning moderate intensity exercise combined with pre-breakfast metformin intake may benefit the management of glycaemia in people with type 2 diabetes.

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

## Additional information

### Data availability statement

Raw data underlying all figures in this manuscript have been deposited publicly https://doi.org/10.6084/m9.figshare.24081417.

### Competing interests

The authors declare they have no competing interests.

### Author contributions

S.P., E.C., F.T. and B.M.G. contributed to the conception or design of the work. B.J.P.C., S.G., A.T., A.K., O.M.D., N.M., M.D., D.B. and B.M.G. contibuted to aquisition, analysis or interpretation of data. All authors contributed to drafting the work or revising it critically for important intelectual content. All authors have read and approved the final version of this manuscript and agree to be accountable for all aspects of the work in ensuring that questions related to the accuracy or integrity of any part of the work are appropriately investigated and resolved. All persons designated as authors qualify for authorship, and all those who qualify for authorship are listed.

### Funding

This study was funded by a European Foundation for the Study of Diabetes/Lilly Young Investigator Award and an NHS Grampian Endowment Research Grant, both to B.M.G. B.M.G. was also supported by a fellowship from the Novo Nordisk Foundation (NNF19OC0055072). B.J.P.C. was supported by a Mexican Government CONAHCyT PhD Studentship (CVU: 516989).

### Acknowledgements

We would like to thank Amanda Cardy from the NRS Primary Care Network for her help in recruiting participants. We would also like to thank Dr Graham Horgan of Biomathematics & Statistics Scotland (BIOSS) for his support with statistical analysis.

### Keywords

chronobiology, chrono-medicine, circadian, exercise, lifestyle, metformin, physical activity, time-of-day, walking

## Supporting information

Additional supporting information can be found online in the Supporting Information section at the end of the HTML view of the article. Supporting information files available:

**Peer Review History**
**Supporting Information**

