## [Peer Review History · The Journal of Physiology]

Morning exercise and pre-breakfast metformin interact to reduce glycaemia in people with Type 2 Diabetes: a randomized crossover trial

Brenda J. Peña-Carrillo, Emily Cope, Sati Gurel, Andres Traslosheros, Amber Kenny, Oscar Michot-Duval, Nimesh Mody, Mirela Delibegovic, Sam Philip, Frank Thies, Dimitra Blana, and Brendan M Gabriel
DOI: 10.1113/JP285722

Corresponding author(s): Brendan Gabriel (brendan.gabriel1@abdn.ac.uk)

The following individual(s) involved in review of this submission have agreed to reveal their identity: Seth Creasy (Referee #1); Amy Keller (Referee #2)

Review Timeline:

Submission Date:	25-Sep-2023
Editorial Decision:	09-Nov-2023
Revision Received:	07-Feb-2024
Accepted:	29-Feb-2024

Senior Editor: Karyn Hamilton

Reviewing Editor: Josiane Broussard

Transaction Report:

Dear Dr Gabriel,

Re: JP-RP-2023-285722 "Morning exercise and pre-breakfast metformin interact to reduce glycaemia in people with Type 2 Diabetes: a randomized crossover trial" by Brenda J Pena Carrillo, Emily Cope, Sati Gurel, Andres Traslosheros, Amber Kenny, Oscar Michot-Duval, Nimesh Mody, Mirela Delibegovic, Sam Philip, Frank Thies, Dimitra Blana, and Brendan M Gabriel

Thank you for submitting your manuscript to The Journal of Physiology. It has been assessed by a Reviewing Editor and by 2 expert referees and we are pleased to tell you that it is potentially acceptable for publication following satisfactory major revision.

LANGUAGE EDITING AND SUPPORT FOR PUBLICATION: If you would like help with English language editing, or other article preparation support, Wiley Editing Services offers expert help, including English Language Editing, as well as translation, manuscript formatting, and figure formatting at www.wileyauthors.com/eoo/preparation. You can also find resources for Preparing Your Article for general guidance about writing and preparing your manuscript at www.wileyauthors.com/eoo/prepresources.

REVISION CHECKLIST:

Please upload two versions of your manuscript text: one with all relevant changes highlighted and one clean version with no changes tracked. The manuscript file should include all tables and figure legends, but each figure/graph should be uploaded as separate, high-resolution files. The journal is now integrated with Wiley's Image Checking service. For further details,

see: <https://www.wiley.com/en-us/network/publishing/research-publishing/trending-stories/upholding-image-integrity-wileys-image-screening-service>

We look forward to receiving your revised submission.

Yours sincerely,

Karyn Hamilton
Senior Editor
The Journal of Physiology

EDITOR COMMENTS

Reviewing Editor:

This is an interesting paper that adds to the literature regarding exercise timing. Reviewers brought up a number of important concerns that need to be addressed, including the actual timing that exercise was completed, the potential impact of exercise on meal timing, and use of Garmin for exercise adherence and sleep outcomes. Authors must also discuss the small sample and clarify statistical comparisons that make the paper difficult to follow.

Senior Editor:

Thank you for submitting your original research manuscript for consideration in a Special Issue of the Journal of Physiology. As part of the review process, we recruited two referees with expertise in this field of study. They both had enthusiasm for your study and for it could potentially contribute to current knowledge regarding exercise timing. However, both referees also raised some serious concerns including (but not limited to) a small sample, some methodology that may lack rigor, the many statistical comparisons impairing the clarity of the results and interpretation, and lack clear communication establishing the innovation of the discoveries made. We would like to invite you to respond directly, point-by-point, to each of the reviewer's concerns, with appropriate major revisions to your manuscript. Please also revisit the statistics policy for The Journal of Physiology and and make appropriate revisions to be in compliance. Revisions, of course, do not guarantee acceptance. Thank you again and we look forward to seeing your revised manuscript if you choose to go this direction.

Statistics Policy note:

The Journal's statistics policy requires use of standard deviation rather than standard error of the mean.

REFeree COMMENTS

Referee #1:

Comments to the Authors: I commend the authors on this interesting study. I believe this would make a nice contribution to the exercise timing literature. However, there are several limitations to that need to be addressed. See below.

Major Comments:

1. In abstract- there is reference to acute glucose AUC being lower after morning exercise but acute is not defined? How long is acute?
2. In abstract - primary and secondary outcomes should be reported even if null. I would favor seeing those results prior to the metformin timing results.
3. Use of Garmin to report exercise intervention adherence. Can the authors provide some validation references around the Garmin Vivosmart 4 to give confidence that it was measuring exercise adherence accurately?
4. Use of Garmin for sleep outcomes. Can the authors provide some validation references around the Garmin Vivosmart 4 to give confidence that it was measuring sleep outcomes accurately?

5. Was meal timing altered by the exercise conditions? Please report and comment on how this may affect results.
6. Primary outcome- in the analysis section it says the study was powered to detect a difference in mean glucose AUC. Which timepoint of was being compared (week 5 and 6)?
7. Did participants eat snacks? How was it ensured that participants did not consume food or caloric intake outside of defined times?
8. Did evening exercise occur prior to or after dinner? Did morning exercise occur prior to or after breakfast? Was this controlled?
9. With the metformin prior to breakfast group was there a specific time of day where glucose was noticeably lower compared to the other conditions?
10. Can authors provide information about what time of day sessions were actually completed? Was this consistent within participants?

Referee #2:

Please see attached.

REQUIREMENTS

- You must start the Methods section with a paragraph headed Ethical Approval. If experiments were conducted on humans, confirmation that informed consent was obtained, preferably in writing, that the studies conformed to the standards set by the latest revision of the Declaration of Helsinki and that the procedures were approved by a properly constituted ethics committee, which should be named, must be included in the article file. If the research study was registered (clause 35 of the Declaration of Helsinki), the registration database should be indicated, otherwise the lack of registration should be noted as an exception (e.g. The study conformed to the standards set by the Declaration of Helsinki, except for registration in a database). For further information see: <https://physoc.onlinelibrary.wiley.com/hub/human-experiments>.

- Papers must comply with the Statistics Policy: https://jp.msubmit.net/cgi-bin/main.plex?form_type=display_requirements#statistics.

In summary:

- If $n \leq 30$, all data points must be plotted in the figure in a way that reveals their range and distribution. A bar graph with data points overlaid, a box and whisker plot or a violin plot (preferably with data points included) are acceptable formats.
- If $n > 30$, then the entire raw dataset must be made available either as supporting information, or hosted on a not-for-profit repository, e.g. FigShare, with access details provided in the manuscript.
- 'n' clearly defined (e.g. x cells from y slices in z animals) in the Methods. Authors should be mindful of pseudoreplication.
- All relevant 'n' values must be clearly stated in the main text, figures and tables, and the Statistical Summary Document (required upon revision).
- The most appropriate summary statistic (e.g. mean or median and standard deviation) must be used. Standard Error of the Mean (SEM) alone is not permitted.
- Exact p values must be stated. Authors must not use 'greater than' or 'less than'. Exact p values must be stated to three significant figures even when 'no statistical significance' is claimed.
- Statistics Summary Document completed appropriately upon revision.

END OF COMMENTS

Confidential Review

25-Sep-2023

Morning exercise and pre-breakfast metformin interact to reduce glycaemia in people with Type 2 Diabetes: a randomized crossover trial

Pena Carrillo, B. et al.

General comments: This crossover clinical trial investigates the benefits, or lack thereof, of exercise timing (morning or evening) in those with type 2 diabetes (T2DM) taking only metformin. Participants, both male and female, exercised in the morning or afternoon/evening following a several week washout period. Those that exercised in the morning showed a significant glucose-lowering effect of exercise as compared to those exercising later in the day. Interestingly, timing of metformin intake (either pre or post breakfast), potentiated this response only in the pre-breakfast, morning exercisers. Also fascinating were data showing significant seasonal differences with blood glucose parameters. Main critiques include omissions of key insights, general organization of text, and under-powered number of participants. This is acknowledged in the limitations section of the Discussion and unfortunately precludes any real insight into sex differences. This is too bad, as equal number of men and women participated. Despite this issue, I find this study robust enough to meaningfully add to a controversial aspect of exercise science.

Key points summary:

- Under the “key question” heading, please add brief experimental details so that readers are ready for the findings.

Introduction:

- Citations from throughout the section should also appear in the paragraph, lines 111-120.
- The summary of the paper in lines 137-138 needs to be more descriptive, or consider omitting. The paper cited in lines 139-140 is unnecessary.
- The sentence in lines 150-151 does not belong here and might be better in the above paragraph. This paragraph discusses metformin, not diurnal exercise.
- Consider having discussion of skeletal muscle be its own paragraph instead of separated into exercise and metformin separately.
- In Figure 1 and 2, please add in those taking metformin pre or post breakfast. I realize this was a secondary outcome, but it is a huge part of your data. Labeling the diagrams up front will help readers be clear.

Methods:

- Line 192 and/or 253-4, please add more detail about the chronotype assessment.
- Line 275, please add the n for metformin pre/post breakfast and pre/post dinner. The dinner data is only mentioned in line 394, so perhaps consider removing mention of it.
- 1.5 Data analysis- please report how you incorporated paired analysis. Also, how did you analyze data for metformin results, as this would be a subset of your total data?
- Please include Supplemental Table 1 in your main manuscript showing no baseline physiological differences between sexes.
- Please also include Supplemental Table 2 in your main manuscript showing seasonal differences. This is interesting and worth mentioning.
- Somewhere in lines 390-396, please clarify that these data are across the whole study. (Figure 8)

Results and figures:

- Some of the figures are bogged down with long X-axis labels. Please consider streamlining them with one legend and shaded bars, or something similar.
- Lines 360-361, are your data switched between summer and winter values?
- Line 337, please include data for evening exercise for comparison. You have it for baseline and morning exercise.
- .

Discussion/Conclusions:

- Please comment on why you had a significant discrepancy between sleep reported with the Garmin and the MCTQ.
- Lines 319-321, you report a trend for a difference in REM sleep between baseline, morning and evening exercise ($p=0.06$). This seems worth a brief discussion, as plenty of studies have shown impacts of exercise on sleep.
- Why do you think the later study period of 5-6 is a driver of the pre-breakfast metformin effect? (Figures 8 and 9) Could acclimation to exercise be a factor?
- Please include the recommended timing of metformin dosage to place your data in context.
- One interpretation of your data completely missing from this section is the potentiation of pre-breakfast metformin on the benefits of fasted exercise on glucose tolerance in the T2DM population. Could this explain your observations? There is tons of literature on this.
- The seasonal differences in physiology are intriguing. With what we know about seasonal and sunlight impacts on circadian rhythms, could this be a cool future

direction. Supplemental Figure 5 shows no significant differences, but there are visual differences between AUC for winter and summer. Could this be due to sample size?

- Please delete lines 433-435. This is repetitive.
- The literature described in lines 436-438 do not align with your observations. Why not?

We thank the reviewers for their time and expertise they have dedicated to these thorough reviews. We have responded specifically to all points below in red text and highlighted changes to the manuscript with a Blue Highlight format. We have also submitted a 'clean' version of the revised manuscript without changes highlighted.

REFEREE COMMENTS

Referee #1:

1. In abstract- there is reference to acute glucose AUC being lower after morning exercise but acute is not defined? How long is acute?

RESPONSE: Acute AUC is AUC for 24 hours. we have amended the abstract to clarify this point from:

Acute glucose area under the curve (AUC)

To

Acute (24 Hour) glucose area under the curve (AUC)

2. In abstract - primary and secondary outcomes should be reported even if null. I would favor seeing those results prior to the metformin timing results.

RESPONSE: We agree with the reviewer and have added 'Primary/Secondary Outcome' labels to specify which data are which. We have also added a sentence to clarify that there were no time-specific differences when data were analysed with a 2-way ANOVA.

3. Use of Garmin to report exercise intervention adherence. Can the authors provide some validation references around the Garmin Vivosmart 4 to give confidence that it was measuring exercise adherence accurately?

4. Use of Garmin for sleep outcomes. Can the authors provide some validation references around the Garmin Vivosmart 4 to give confidence that it was measuring sleep outcomes accurately?

RESPONSE: Regarding points 3-4. We have change section 1.4.1 from:

Physical activity data (assessed as step count per day and heart rate) as well as sleep quality were collected via a wrist-based physical activity monitor Garmin Vivosmart 4 (Garmin Ltd, Olathe, KS, US),

To

Physical activity data (assessed as step count per day and heart rate) as well as sleep quality were collected via a wrist-based physical activity monitor Garmin Vivosmart 4 (Garmin Ltd, Olathe, KS, US), this device can accurately measure physical activity variables with a low percentage of error: HR, 5.56 bpm; steps-per-day 36.75 (Teresawa, et al., 2023). This device can also detect changes in sleep variables, especially during long-term monitoring (Mouritzen et al., 2020).

5. Was meal timing altered by the exercise conditions? Please report and comment on how this may affect results.

RESPONSE: We have added meal timings to 2.4 Results section.

Meal timing for dinner remained consistent throughout the trial, with no significant differences observed ($p=0.49$) in dinner times for baseline (18:34±00:32 hour), morning

(18:31±00:40 hour), or evening exercise (18:40±00:53 hour). While breakfast times were significantly later during evening exercise (09:17±00:52 hour) compared to baseline (08:57±00:49 hour) breakfast times were not affected for the morning exercise ($p=0.30$). The breakfast time for the morning arm (09:04±00:38 hour) was not significantly different from the baseline period.

This study was designed to minimize interference with participants' daily activities, allowing them to adhere to their usual meal patterns despite incorporating exercise, to achieve this, participants were instructed to perform exercise within a flexible 3-hour window, either between 7:00-10:00 AM or 4:00-7:00 PM. In addition, the type and intensity of the exercise may not have prompted significant changes in appetite or meal habits. Moderate exercise, for instance, may have less impact on meal timing compared to intense workouts.

6. Primary outcome- in the analysis section it says the study was powered to detect a difference in mean glucose AUC. Which timepoint of was being compared (week 5 and 6)?

RESPONSE: The data that were used to perform the power calculation was based on week 1 of a 2-week exercise study (Savikj, Gabriel et al., 2019) with data plotted as a 24-hour time course with each exercise day used as an internal replicate, giving each participant 3 internal replicates. The previous study had 11 participants.

7. Did participants eat snacks? How was it ensured that participants did not consume food or caloric intake outside of defined times?

RESPONSE: We have added snacks information to 1.4.3 methodology section.

“During the analysis, we divided the dietary intake into Breakfast, lunch, and dinner for comparison of time or size of meal intake. Snacks were combined with the nearest meal by time.”

8. Did evening exercise occur prior to or after dinner? Did morning exercise occur prior to or after breakfast? Was this controlled?

RESPONSE: We have added exercise times to 2.2 results section, breakfast and dinner times have been added to 2.4 results section.

Participants were instructed to perform exercise within a flexible 3-hour window, either between 7:00-10:00 AM or 4:00-7:00 PM, mean evening exercise (16:48±00:40 hour) occurred before dinner time (18:40±00:53 hour); mean morning exercise (08:46 hour±00:54) occurred before breakfast time (09:04±00:38 hour). Participants were not instructed to exercise before or after meals but were instructed to maintain their habitual routines outside of the specified exercise.

9. With the metformin prior to breakfast group was there a specific time of day where glucose was noticeably lower compared to the other conditions?

RESPONSE: We have added data and figure 10 to 2.5 results section.

“In the morning exercise session, a glucose difference ($p=0.03$) was observed at 22:00 hours between metformin before and after breakfast (Figure 10B). No differences were observed at any time point during the 24-hour time-course between metformin before and after breakfast for baseline and evening exercise (Figure 10A,C)”.

10. Can authors provide information about what time of day sessions were actually completed? Was this consistent within participants?

RESPONSE: We have added time of day sessions to 2.2 Results section and supplementary material (Supplementary Figure 2).

Participants showed consistent time of exercise sessions withing window times scheduled. On average, participants completed sessions at 08:46±00:54 hours in the morning and 16:48±00:40 hours in the evening (Supplementary Figure 2).

Referee #2:

Key points summary

Under the “key question” heading, please add brief experimental details so that readers are ready for the findings.

RESPONSE: We have added the following text to summarise the experimental procedures of the study:

“18 participants with type 2 diabetes undergoing metformin monotherapy (age 61±8.2 year, mean±SD) had blood glucose monitored and completed a 16-week crossover trial including 2-week baseline recording, six weeks randomly assigned to a morning exercise (7-10am) or evening exercise (4-7pm), and a two-week wash-out period. “

Introduction:

Citations from throughout the section should also appear in the paragraph, lines 111-120.

RESPONSE: Agreed, we have added the sentence below to include these citations.

“This time-of-day of exercise effect upon blood glucose regulation appears to be supported by other recent studies (Mancilla et al., 2021; Moholdt et al., 2021; van der Velde et al., 2023; Qian et al., 2023).”

The summary of the paper in lines 137-138 needs to be more descriptive or consider omitting.

RESPONSE: We have omitted this citation

The paper cited in lines 139-140 is unnecessary.

RESPONSE: We have omitted this citation

The sentence in lines 150-151 does not belong here and might be better in the above paragraph. This paragraph discusses metformin, not diurnal exercise.

RESPONSE: We feel that metformin intake constitutes a part of the holistic diurnal environment and therefore is an important consideration in the timing of T2D management strategies. We have clarified this by editing the sentence as below, Introduction section.

“One must also consider the timing of various treatment strategies within the holistic diurnal environment in regards to management of type 2 diabetes.”

Consider having discussion of skeletal muscle be its own paragraph instead of separated into exercise and metformin separately.

RESPONSE: Agreed, we have combined sections into a skeletal muscle paragraph. Introduction section.

In Figure 1 and 2, please add in those taking metformin pre or post breakfast. I realize this was a secondary outcome, but it is a huge part of your data. Labeling the diagrams up front will help readers be clear.

RESPONSE: We have changed Figure 1 and 2 to include participant stratification based on metformin intake before/after breakfast. Labelling of figure 1 has been changed to be more descriptive.

Methods:

Line 192 and/or 253-4, please add more detail about the chronotype assessment.

RESPONSE: We have added more details about chronotype assessment to 1.4.1 Physical activity methods section.

“The participants' chronotype, reflecting individual preferences in sleep-wake rhythms, was identified through the self-administered Munich Chronotype Questionnaire, consisting of 29 questions, the scale assesses wake and sleep patterns on both work and free days, energy levels throughout the day, sleep latency and exposure to daylight”.

Line 275, please add the n for metformin pre/post breakfast and pre/post dinner. The dinner data is only mentioned in line 394, so perhaps consider removing mention of it.

RESPONSE: We have added Information about number of participants included for metformin pre/post breakfast and pre/post dinner to 1.4.3 methods section. Dinner data has been added, supplementing the existing metformin pre/post breakfast numbers. We consider it important to keep dinner data and analysis as part of the manuscript.

“As dietary intake and metformin time and dose were self-reported, the data analysis of the food diaries and the metformin analysis before or after breakfast corresponded to 14 participants that fulfilled this requirement. For baseline period, metformin pre-breakfast (n=9), metformin post-breakfast (n=5); Morning exercise, metformin pre-breakfast (n=8), metformin post-breakfast (n=5); Evening exercise, metformin pre-breakfast (n=10), metformin post-breakfast (n=4). Metformin analysis before or after dinner corresponded to 13 participants. For baseline period, metformin pre-dinner (n=10), metformin post-dinner (n=3); Morning exercise, metformin pre-dinner (n=9), metformin post-dinner (n=4); Evening exercise, metformin pre-dinner (n=7), metformin post-dinner (n=6)”.

1.5 Data analysis- please report how you incorporated paired analysis. Also, how did you analyze data for metformin results, as this would be a subset of your total data?

RESPONSE: We have added detail of statistics analyses to 1.5 data analysis section.

“Adherence to exercise protocol variables were compared by means using paired t-test. Baseline characteristics between males and females, winter and summer, metformin before/after breakfast were compared by means using unpaired t-test. Changes in glucose during exercise intervention and caloric intake were analysed by One-way ANOVA or two-way ANOVA followed up by Holm-Šidák's multiple comparisons test. Given that participants recorded time and doses of metformin intake, the AUC glucose levels were categorized according to whether metformin was taken before or after breakfast and dinner, these categories were analysed by unpaired t-test.”

Please include Supplemental Table 1 in your main manuscript showing no baseline physiological differences between sexes.

RESPONSE We have included supplemental Table 1 in 2.3 results section.

Please also include Supplemental Table 2 in your main manuscript showing seasonal differences. This is interesting and worth mentioning.

RESPONSE: We have included supplemental Table 2 in 2.3 results section.

Somewhere in lines 390-396, please clarify that these data are across the whole study. (Figure 8)

RESPONSE: We have attempted to clarify this sentence that the comparisons were across all study periods for the acute AUC Glucose to 2.5 results section.

“Only when the participants performed morning exercise was acute 24 AUC glucose lower ($p=0.01$) in participants taking metformin before breakfast (152.5 ± 29.95 mmol/L) compared with participants taking metformin after breakfast (227.2 ± 61.51 mmol/L) (Figure 8).”

Results and figures:

Some of the figures are bogged down with long X-axis labels. Please consider streamlining them with one legend and shaded bars, or something similar.

RESPONSE: We have changed the labels on the X-axis for figures 6, 7, 8, 9 and Supplemental Figure 6 have been modified.

Lines 360-361, are your data switched between summer and winter values?

RESPONSE: Yes, there was an inadvertent switching of values during the typing, we have now rectified the all the values, 2.3 results section.

“Physiological characteristics of participants showed lower HbA1c levels ($p=0.04$) during summer (55.6 ± 9.3 mmol/mol) compared to winter (70.7 ± 15.5 mmol/mol) (Table 3). There was a trend ($p=0.07$) showing lower AUC glucose levels in summer (169.8 ± 2.1 mmol/L) compared to the winter (222.4 ± 14.3 mmol/L).”

Line 337, please include data for evening exercise for comparison. You have it for baseline and morning exercise.

RESPONSE: We have added values for evening exercise to section 2.3.

“Whereas there were no significant differences ($p=0.12$) in acute glycemia levels during evening exercise (191.7 ± 48.53 mmol/L) compared with baseline (210.3 ± 76.7 mmol/L).”

Discussion/Conclusions:

Please comment on why you had a significant discrepancy between sleep reported with the Garmin and the MCTQ.

RESPONSE: We have added text addressing the discrepancy and also some limitations of our sleep assessment methods below in the discussion section.

“We also note that we used Garmin Vivosmart 4 to monitor sleep duration and quality. As noted above, this may not be as precise for describing sleep architecture as gold-standard techniques such as PSG.

We identified a discrepancy between the self-reported MCTQ and the sleep duration reported by Garmin Vivosmart 4. It is known that retrospective self-reported sleep duration can differ substantially from actigraphy and polysomnography assessments (Matthews et al., 2018). Therefore, this is a likely cause of this discrepancy.”

Lines 319-321, you report a trend for a difference in REM sleep between baseline, morning and evening exercise ($p=0.06$). This seems worth a brief discussion, as plenty of studies have shown impacts of exercise on sleep.

RESPONSE: We have added the text below in the discussion section to address this point.

“Our findings show a trend for sleep duration to be longer during evening exercise periods compared with morning exercise, although neither differed from baseline. While evening exercise has been shown to reduce sleep quality (Saidi et al., 2023) in certain populations, this is not consistent across the literature (Youngstedt et al., 1999; Myllymaki et al., 2011; Frimpong et al., 2021), and may be dependent on chronotype (Saidi et al., 2023). Thus, it is not clear if our finding is due to a lack of statistical power, or whether the null-finding is true. Additionally, evening exercise may impact differently on the sleep of those with an earlier chronotype”.

Why do you think the later study period of 5-6 is a driver of the pre-breakfast metformin effect? (Figures 8 and 9) Could acclimation to exercise be a factor?”

RESPONSE: We have discussed our thoughts on this in the discussion section with the text below:

“In addition to seasonal differences, long-term acclimation to the exercise intervention may be an important factor to consider. We observed significantly lower glucose levels during the morning exercise period in people who took metformin before breakfast compared to after breakfast both acutely (initial 24 hours), and in the final 2 weeks of the morning exercise period. However, we observed only a trend ($p=0.07$) for lower glucose in the first 2 weeks of exercise when participants were stratified in the same manner. It is unclear whether this is due to statistical power, or whether there is an acute and training acclimation effect in regards to the timing interaction between exercise and metformin intake. There have been few studies assessing divergent diurnal outcomes of exercise in people with Type 2 Diabetes over a longer training period. Retrospective studies suggest that even over a longer exercise training period, timing of exercise may be important in glycaemic management (Qian et al., 2023), however these findings should also be tested in specifically designed longitudinal studies.”

Please include the recommended timing of metformin dosage to place your data in context.

RESPONSE: We have added the following text to 3. Discussion section:

“Clinicians often recommend taking metformin with or after meals, including breakfast (NICE, 2023).”

One interpretation of your data completely missing from this section is the potentiation of pre-breakfast metformin on the benefits of fasted exercise on glucose tolerance in the T2DM population. Could this explain your observations? There is tons of literature on this.

RESPONSE: This is a good point, we have added discussion on this with the text below.

“Another possible diurnal phenomenon that may contribute to our observed effects is that of the potentiation of pre-breakfast metformin upon the effect of fasted morning exercise upon blood glucose. The effect of fed vs fasted exercise on glycaemic control has been debated in the literature with differing outcomes dependent on experimental conditions and the populations studied (Wallis & Gonzalez, 2019). In our study, we did not specify that participants should perform exercise before or after breakfast, the mean time of morning exercise was 08:46±00:54, while breakfast was 09:04±00:38 during the morning exercise

period. Therefore, our data suggest it is possible that pre-breakfast metformin does have a potentiation effect on fasted morning exercise.”

The seasonal differences in physiology are intriguing. With what we know about seasonal and sunlight impacts on circadian rhythms, could this be a cool future direction. Supplemental Figure 5 shows no significant differences, but there are visual differences between AUC for winter and summer. Could this be due to sample size?

RESPONSE: Thanks for this observation, we have discussed by adding the text below.

“In this study we recruited two cohorts in the summer and winter to account for seasonal differences in our observations. We were thus able to stratify participants into winter and summer cohorts, however the study was not designed to detect primary outcomes in this manner. There were no statistically significant differences between winter and summer cohorts, although the winter cohort did have consistently higher AUC glucose levels in all periods of the trial. An observational study performed in a Canadian healthcare setting found that hospital encounters for hyperglycaemia peaked in January compared with the other calendar months (11% increase) (Clemens et al., 2017). Although seasonal differences in glycaemic regulation are clearly multifactorial, optimisation of management strategies for Type 2 Diabetes should also consider seasonal variations.”

Please delete lines 433-435. This is repetitive.

RESPONSE: We have deleted these.

The literature described in lines 436-438 do not align with your observations. Why not?

RESPONSE: Given these studies used the standard clinical recommended timing of metformin, we feel that the newly added preceding sentence (below) now clarifies our thoughts on these lines: i.e. that optimising timing of metformin alongside meals and exercise may somewhat overcome the interference effects between metformin and exercise observed in the literature.

“Clinicians often recommend taking metformin with or after meals, including breakfast (e.g. National Institute for Health and Care Excellence guidelines, Nov 2023).”

Dear Dr Gabriel,

Re: JP-RP-2024-285722R1 "Morning exercise and pre-breakfast metformin interact to reduce glycaemia in people with Type 2 Diabetes: a randomized crossover trial" by Brenda J. Peña-Carrillo, Emily Cope, Sati Gurel, Andres Traslosheros, Amber Kenny, Oscar Michot-Duval, Nimesh Mody, Mirela Delibegovic, Sam Philip, Frank Thies, Dimitra Blana, and Brendan M Gabriel

We are pleased to tell you that your paper has been accepted for publication in The Journal of Physiology.

Authors should note that it is too late at this point to offer corrections prior to proofing. Major corrections at proof stage, such as changes to figures, will be referred to the Editors for approval before they can be incorporated. Only minor changes, such as to style and consistency, should be made at proof stage. Changes that need to be made after proof stage will usually require a formal correction notice.

If you would like to receive our 'Research Roundup', a monthly newsletter highlighting the cutting-edge research published in The Physiological Society's family of journals (The Journal of Physiology, Experimental Physiology and Physiological Reports), please click this link, fill in your name and email address and select 'Research Roundup': <https://www.physoc.org/journals-and-media/membernews/>.

Yours sincerely,

Karyn Hamilton
Senior Editor
The Journal of Physiology

P.S. - You can help your research get the attention it deserves! Check out Wiley's free Promotion Guide for best-practice recommendations for promoting your work at www.wileyauthors.com/eeo/guide. You can learn more about Wiley Editing Services which offers professional video, design, and writing services to create shareable video abstracts, infographics, conference posters, lay summaries, and research news stories for your research at www.wileyauthors.com/eeo/promotion.

IMPORTANT NOTICE ABOUT OPEN ACCESS: To assist authors whose funding agencies mandate public access to published research findings sooner than 12 months after publication, The Journal of Physiology allows authors to pay an Open Access (OA) fee to have their papers made freely available immediately on publication.

You can check if your funder or institution has a Wiley Open Access Account here: <https://authorservices.wiley.com/author-resources/Journal-Authors/licensing-and-open-access/open-access/author-compliance-tool.html>.

EDITOR COMMENTS

Reviewing Editor:

Both reviewers thank and commend the authors on a thorough revision and have no further comments or suggestions.

Senior Editor:

Thank you for your detailed responses to the peer reviewer critiques with careful revisions to the manuscript. We are pleased to accept this manuscript for publication. Congratulations and thank you for your interest in The Journal!

REFEREE COMMENTS

Referee #1:

I commend the authors on their thorough revision. I have no further comments or suggestions.

Referee #2:

The authors have done a thorough job in addressing the reviewer comments, strengthening the manuscript quite a bit. I recommend publishing.

1st Confidential Review

07-Feb-2024